# The Influence of Humidity on the Performance of a Low-Cost Air Particle Mass Sensor and the Effect of Atmospheric Fog

Rohan Jayaratne[1], Xiaoting Liu[1], Phong Thai[1], Matthew Dunbabin[2], Lidia Morawska [1]

[1] International Laboratory for Air Quality and Health, Queensland University of Technology, Brisbane, QLD 4001, Australia
[2] Institute for Future Environments, Queensland University of Technology, Brisbane, QLD 4001, Australia

*Correspondence to*: Lidia Morawska (l.morawska@qut.edu.au)

**Abstract.** While low-cost particle sensors are being increasingly used in numerous applications, most of them have no heater or dryer at the inlet to remove water from the sample before measurement. Deliquescent growth of particles and the formation of fog droplets in the atmosphere can lead to significant increases in particle number concentration (PNC) and mass concentrations reported by such sensors. We carried out a detailed study using a Plantower PMS1003 low-cost particle sensor, both in the laboratory and under actual ambient field conditions, to investigate its response to increasing humidity and the presence of fog in the air. We found significant increases in particle number and mass concentrations at relative humidity above about 75%. During a period of fog, the total PNC increased by 28%, while the PNC larger than 2.5 µm increased by over 50%. The $PM_{10}$ concentration reported by the PMS1003 was 46% greater than that on the standard monitor with a charcoal dryer at the inlet. While there is a causal link between particle pollution and adverse health effects, the presence of water on the particles is not harmful to humans. Therefore, air quality standards for particles are specifically limited to solid particles and standard particle monitoring instruments are fitted with a heater or dryer at the inlet to remove all liquid material from the sample before the concentrations are measured. This study shows that it is important to understand that the results provided by low-cost particle sensors, such as the PMS1003, cannot be used to ascertain if air quality standards are being met.

## 1 Introduction

The rapid technological advancements in the fields of material science, digital electronics and wireless communication have given rise to a wide range of low-cost air quality sensors that are now readily available on the market. These sensors are increasingly being used in many applications that were previously not achievable with conventional expensive equipment (Kumar et al., 2015; Rai et al., 2017; Snyder et al., 2013). Some of these applications are the monitoring of personal exposure and indoor air pollution and the gathering of high-resolution spatiotemporal air pollution data by means of extensive sensor networks. The data thus derived are being utilised for a variety of air pollution management tasks such as supplementing conventional air pollution monitoring, understanding the link between pollutant exposure and human health, emergency response management, hazardous leak detection and source compliance monitoring. In the process, they also

serve to increase the community's awareness and engagement towards air quality issues (Snyder et al., 2013; Jovasevic-Stojanovic et al., 2015; Rai et al., 2017).

However, there are many questions regarding the reliability and, in particular, the accuracy of these low-cost sensors and their suitability in the applications that they are being used (Lewis and Edwards, 2016). Many of these sensors have serious limitations. For example, while many particle sensors respond well to high concentrations, they fail to do so at lower levels such as typical ambient concentrations (Jayaratne et al., 2018; Rai et al., 2017; Kelly et al., 2017). Single gas sensors are very often affected by other interfering gases (Fine et al., 2010; Piedrahita et al., 2014), while environmental parameters such as temperature and humidity can also affect the performance of these sensors under certain conditions (Holstius et al., 2014; Rai et al., 2017; Crilley et al., 2018; Jayaratne et al., 2018).

In this paper, we investigate the effect of atmospheric relative humidity on the performance of a low-cost particulate matter sensor. Humid conditions can affect the performance of a sensor in several ways. For example, sensors that operate on the principle of light scattering are affected as the particle refractive indices are dependent on relative humidity (Hänel, 1972; Hegg et al., 1993). High humidity can cause condensation to form on electrical components leading to resistive bridges across components. In gas sensors, condensation on the sensor surfaces can affect the reactions that give rise to the measurable electric currents.

Hygroscopic growth occurs when the relative humidity exceeds the deliquescence point of a substance. There are many hygroscopic salts such as sodium chloride, that absorb water and grow at relative humidity as low as 70%, present in the atmosphere, especially in marine environments (Hu et al., 2010). Jamriska et al. (2008) found a significant effect of relative humidity on traffic emission particles in the size range 150-880 nm and attributed it to hygroscopic particle growth. Crilley et al. (2018) demonstrated a significantly large positive artefact in measured particle mass by an Alphasense OPC-N2 sensor during times of high ambient relative humidity. Manikonda et al. (2016) cautioned against using PM sensors in outdoor locations at high humidity due to hygroscopic growth of particles. In circumstances where the relative humidity approaches 100%, there is the possibility of mist or fog droplets that are detected as particles. While there is a causal link between particle pollution and adverse human health effects, the presence of water on the particles play no part in it. Therefore, air quality standards for particles are based on the dry, solid material only, and stipulate that the liquid portion must be eliminated when measuring particle mass for regulatory purposes. In order to achieve this, many conventional particle mass monitors such as the standard tapered element oscillating microbalance (TEOM) employ a charcoal heater at its inlet to remove all liquids from the particles that are being measured (Charron et al., 2004; Alexandrova et al., 2003). Thus, sensors with no drying facility at the inlet measure what is actually present in the environment rather than what is required under regulatory protocols.

The composition of particles in the atmosphere of Brisbane, as derived from Harrison (2007) is shown in Fig. 1. The subtropical, near-coastal environment is characterised by the presence of several hygroscopic salts such as sodium chloride, ammonium sulphate and ammonium nitrate that have deliquescence relative humidities in the range of 70% to 80% (Hu et al., 2010). Many particles in the air in Brisbane contain these salts in varying concentrations. Once the relative humidity

exceed the respective deliquescence values, those salts begin to absorb water, resulting in particle growth and the excess water is registered by PM sensors, unless they are removed at the instrument inlets by heating or drying (Alexandrova et al., 2003). While more expensive instruments such as the TEOM have built-in drying features at the sample inlets, it is not standard on low-cost sensors and even in many other mid-cost monitors such as the TSI DustTrak (Kingham et al., 2006).

There have been very few studies of the effect of relative humidity on the performance of low cost sensors. Wang et al. (2015) investigated the performance of three low cost particle sensors based on light scattering and concluded that the absorption of infrared radiation by a film of water on a particle can cause an overestimation of the derived particle mass concentration due to the reduced intensity of light received by the phototransistor. Hojaiji et al. (2017) showed that the particle mass concentration reported by a Sharp PM sensor increased when the humidity was increasing but not when it was

decreasing. While several studies have drawn attention to a possible effect of humidity on the performance of low cost sensors, no study has reliably quantified the effect. This study was carried out to investigate and to assess the magnitude of the effect of relative humidity on the performance of a low-cost particle sensor and to understand the mechanisms involved.

## 2 Method

In this study, we focussed on the effect of relative humidity on the performance of a low-cost particle sensor in the

laboratory and under real world conditions in an outdoor location at an air quality monitoring station with standard instrumentation.

### 2.1 The Test Sensor

Prior to commencing this study we tested a range of low-cost particle sensors, including the Sharp GP2Y, Shinyei PPD42NS, Plantower PMS1003, Innociple PSM305 and the Nova SDS011 (Jayaratne et al., 2018). All of them were found

to be affected to some degree by humidity with the Sharp and Shinyei being affected at relative humidity as low as 50% while the other three showed deviations from the standard instruments when the relative humidity exceeded 75-80%. Considering their performance characteristics, the Plantower PMS1003 was selected as the most suitable sensor for this study. This sensor was selected because it is freely available, low-cost (around US$20) and its performance characteristics have been previously investigated extensively in our laboratories and found to be superior to the other sensors tested

(Jayaratne et al, 2018). The PMS1003 is a compact particle sensor that monitors particles larger than 0.3 µm in diameter. It operates by drawing the sample air using a miniature fan into a small inbuilt chamber, where the particles are exposed to a fine laser beam. The scattered light is detected by a photodetector which produces an electrical output. The signal is processed using a complex algorithm to provide real-time readings of particle mass concentration in three ranges – $PM_1$, $PM_{2.5}$ and $PM_{10}$, together with particle number concentrations (PNC) in six size ranges – greater than 0.3, 0.5, 1.0, 2.5, 5 and

10 µm, at intervals down to 2s. All three PM values are reported in units of µg m$^{-3}$, while the PNCs are reported as per 0.1L or dL$^{-1}$.

The PMS1003 was mounted on a custom interface board including a low-power microcontroller with multiple serial interfaces, a high-resolution 16-bit analog to digital converter and a real-time clock that provided accurate time-stamping of the measurements. The PMS1003 was attached to a frame along with the interface board, allowing unobstructed airflow into and out of the device. The microcontroller was programmed to perform the necessary signal processing and power management. The time-stamped data were transferred in real-time via USB serial communications to a computer and logged into a text file for post-analysis.

## 2.2 Standard Instrumentation

In the laboratory experiments, we used a TSI 8530 DustTrak DRX aerosol monitor with a $PM_{2.5}$ impactor. The instrument has an inbuilt data logger. The sample air is drawn through the inlet which has no drying facility to remove the liquid portions of the particles, if any. Prior to the study, the DustTrak was calibrated against a standard TEOM in the laboratory. With dry ambient aerosols, the $PM_{2.5}$ concentrations reported by the two instruments agreed to within 10% (Jayaratne et al., 2018). With normal ambient aerosols, the readings again agreed closely until the relative humidity exceeded about 75% when the DustTrak readings were significantly greater than that of the TEOM. The air quality monitoring station, where the field study was conducted, contained two TEOMs providing accurate 5-min readings of $PM_{2.5}$ and $PM_{10}$, together with accurate measurements of air temperature and relative humidity.

The station also included a nephelometer to monitor atmospheric visibility in terms of the particle back-scatter (BSP) coefficient, reported in units of $Mm^{-1}$. The BSP corresponds to the concentration of particles in the air and provides an estimate of the visibility. Observations have shown that its value typically ranges from about 5-15 $Mm^{-1}$ on a 'clean' day to about 50 $Mm^{-1}$ on polluted days with, for example, traces of smoke in the atmosphere. However, during periods of fog, the value is generally much higher. Careful visual observations over a period of several weeks in Brisbane confirmed that the presence of mist or fog in the air generally resulted in BSP readings greater than 100 $Mm^{-1}$. Where visual observations were not possible, such as during the night, this value of BSP was used in this study as an indicator of fog in the atmosphere.

## 2.3 Laboratory Experiments

The laboratory experiments were carried out in a 1 $m^3$ chamber. Ambient air from outside the building was drawn into the chamber by means of a low power air pump at a flow rate of about 1 L $min^{-1}$ so that the particle concentration in the chamber was maintained at a relatively steady value close to that of the outdoor air. The interface board with the PMS1003 was placed on a raised platform inside the chamber and directly connected to the computer which was placed outside. Readings were obtained in real-time at intervals of 5 s. The DustTrak monitor was located outside the chamber, sampling the air through a short length of conductive rubber tubing. A small fan on the floor of the chamber was used to ensure that the air was well mixed to give uniform particle concentrations throughout its volume. The humidity in the chamber was increased by introducing moist tissue paper. The relative humidity was monitored with a TSI 7545 Indoor Air Quality meter.

**2.4 Field Experiments**

The field measurements were carried out at an air quality monitoring station, situated close to a busy road, carrying approximately 100 vehicles per min during the day. The PMS1003 was housed in a sealed weather-proof box of dimensions 150x120x100 mm, and the built-in fan was used to draw ambient air from the outside through an aperture in the box.

Readings were obtained at 5 min intervals over a continuous period of 24 days between 21st July and 14th August 2017.

**3 Results**

**3.1 Laboratory Experiments**

With the steady introduction of ambient air, the $PM_{2.5}$ concentration in the chamber was maintained at about $10 \pm 1$ µg m$^{-3}$. PNCs were typically about 1000 and 50 dL$^{-1}$ in the size bins larger than 0.3 and 1.0 µm, respectively. As the humidity in the

chamber was gradually increased, the particle mass concentrations reported by the PMS1003 did not show a significant change until the relative humidity reached about 78%. Fig. 2 shows the corresponding $PM_{2.5}$ concentrations reported by the PMS1003 and the DustTrak. The critical relative humidity beyond which the $PM_{2.5}$ concentration reported by the PMS1003 begins to deviate from the previous ambient value is indicated by the broken line in the figure. Beyond this value, the $PM_{2.5}$ readings indicated by the PMS1003 increased steadily from about 9 µg m$^{-3}$ at a relative humidity of 78% to about 16 µg m$^{-3}$

at the maximum relative humidity of  89% achieved in this experiment – an increase of almost 80%. Interestingly, the corresponding increase in the number concentration of particles in the smallest size bin, 0.3 to 0.5 µm, was of the order of 10%, suggesting that the increase in $PM_{2.5}$ was mainly as a result of particle growth by water absorption and not due to the formation of new water droplets. Thereafter, gradually allowing the relative humidity to decrease resulted in a hysteresis effect with no significant reduction in $PM_{2.5}$ concentration until the relative humidity had decreased to about 50%. The

DustTrak aerosol monitor also showed a similar trend, with no change in $PM_{2.5}$ concentration reading until the relative humidity exceeded about 75% and then a steady increase in concentration as the humidity was increased further (Fig. 2).

As observed in the figure, the $PM_{2.5}$ readings of the particular PMS1003 sensor used in this experiment were consistently higher than the readings on the DustTrak. In general, the readings of the PMS1003 sensors differed between the individual units. The differences depended on the type of aerosol and the concentration being measured. At the low concentrations

found in the ambient environment of Brisbane, the coefficient of variation between 'identical' PMS1003 sensor units was about 0.07, as reported in detail in Jayaratne et al. (2018).

**3.2 Field Experiments**

Fig 3 shows the time series of the $PM_{2.5}$ concentrations reported by the PMS1003 and the standard TEOM during the entire

duration of the study. Also shown is the relative humidity during this period. The relative humidity exhibited a daily cycle with a minimum in the early afternoon and a maximum at night. Note that the peak $PM_{2.5}$ concentrations indicated by both

instruments generally coincided with the time when the relative humidity reached its maximum value near dawn each day. The maximum value often coincided with episodes of fog, although its value did not reach 100%. It is likely that this was a consequence of a limitation of the instrument. At such times, the PMS1003 reading was generally higher than the TEOM. However, from Fig 3, it is observed that on many days, the readings on both instruments increased during times when the relative humidity was high, suggestive that the TEOM did not remove all of the liquid portion of the aerosols. In the afternoon, the TEOM reading was often higher than the PMS1003. This is probably because most of the aerosols in the atmosphere at this time were ultrafine particles from motor vehicle emissions. The size of these particles are below the minimum detectable size limit of the PMS1003 which is 0.3 µm.

Fig. 4 shows the hourly $PM_{2.5}$ concentrations reported by the PMS1003 and TEOM on the night of the 6-7 August, which was realtively humid at the air quality monitoring station. On this night, the relative humidity reported by the monitoring station increased steadily through the night from 76% at 18:00 h, exceeding 90% at 5:00 h the next morning. Fog was visually observed at the site during the early morning hours. The TEOM showed little variation in $PM_{2.5}$ concentration over this period but the value reported by the PMS1003 increased sharply and doubled by the morning.

The PNC values reported by the PMS1003 in all size bins were also higher during periods of fog. Under stable conditions, the PNCs reported by the PMS1003 in the various size bins are generally linearly related. In Fig 5, we show the number concentration of particles larger than 1.0 µm against the corresponding number in the lowest size bin, 0.3 to 0.5 µm on the 31$^{st}$ August when there was an episode of fog visually observed during the early morning. The points under the broken line in the graph correspond to the day time and the first half of the night when there was no fog observed. A linear relationship is evident at this time as illustrated by the straight line in Fig. 5. However, there is a departure from this trend in the section of the graph above the broken line which coincides with the period when the relative humidity was above 75%. As indicated, the points at the upper end of this graph correspond to the early morning hours during the presence of fog, clearly suggesting that the PMS1003 detects water droplets in the air.

Next, we compare the $PM_{2.5}$ concentration reported by the PMS1003 and TEOM during a day with no fog and on a day with an episode of fog (Fig 6). Fig. 6(a) shows the results on the 24$^{th}$ July when the relative humidity did not exceed 80% and there were no visual reports of fog. The concentrations shown by both instruments remained below 20 µg m$^{-3}$ during much of the day and never exceeded 30 µg m$^{-3}$ at any time. Fig. 5(b) is the corresponding graph for the 30$^{th}$ August when there was fog observed between 3:00 and 06:30 AM. During the morning, the indicated relative humidity touched 100% at 3:00 AM and decreased to 90% soon after the fog dispersed at about 6:30 AM. The PMS1003 showed a sharp increase in $PM_{2.5}$ concentration, almost doubling from midnight to 6:30 AM, while the TEOM did not show a significant increase during this time period. Thereafter, the concentrations reported by both instruments showed a steady decline and attained agreement at about 9:00 AM.

Fig 7 shows the corresponding PNCs reported by the PMS1003 at 3.00, 6.00, 9.00 and 12.00 h on the day shown in Fig 6(b). The bars represent the particle number dL$^{-1}$ at all sizes greater than the values given in the legend in µm. For example, we see approximately 1000 particles that are larger than 0.5 µm in 1 dL at 3;00 AM. Note that the fog first became evident at

3:00 AM and dissipated by 6:30 AM. The relative humidity and $PM_{2.5}$ concentrations reported by the PMS1003 and TEOM at the four times are given below the figure. During the time of fog, the total PNC increased by 28%, while the PNC larger than 2.5 µm increased by over 50%. Considering the particle mass in the air, the TEOM showed a $PM_{10}$ concentration increase of about 31% while the PMS1003 showed a significantly larger increase of 46%. All these observations indicate a
moderate increase in the number of fog droplets in the air, accompanied by a very strong rate of hygroscopic mass growth.

## 4 Discussion and Conclusion

It is well known that humid air can have a negative effect on the performance of electronic circuits. For example, moisture in the air can decrease the insulation resistance in electrolytic capacitors and increase the leakage currents in transistors and integrated circuits, reducing the gain. In our previous tests (Jayaratne et al., 2018), we showed that the performance of some
low-cost particle sensors such as the Sharp GP2Y and the Shinyei PPD42NS were affected at relative humidity as low as 50%. The adverse effect was a fluctuation of the output signals, rather than a steady increase with humidity. This was obviously not due to particle growth, and we conclude that the electronics or optical characteristics were, in some way, responsible for these effects.

However, sensors such as the Plantower PMS1003, Innociple PSM305 and the Nova SDS011, as well as particle monitors such as the TSI DustTrak, did not show a marked effect until the relative humidity exceeded about 75%, when they began to show a steady increase. The results of the present study, with the PMS1003 and the DustTrak showed that this was due to particle growth. When the relative humidity is high, particle growth and fog are detected and reported by particle monitoring instruments that do not have drying facilities at the sample inlets. This effect needs to be taken into consideration when using
low-cost particle sensors, especially in environments that contain hygroscopic salts such as near coastal regions. Particles in the air begin to grow once the deliquescence relative humidity is exceeded. For example, two hygroscopic salts that are commonly found in Brisbane air are sodium chloride and ammonium sulphate. These have deliquescence points of approximately 74% and 79% respectively (Hu et al., 2010;Wise et al., 2007). Aerosol particles that contain these substances will absorb moisture and grow when the relative humidity exceeds these values. Our observations are in good agreement
with these studies. The high $PM_{2.5}$ concentration values reported by the PMS1003 during the early morning hours in Fig 6(b) are due to hygroscopic growth of particles followed by the formation of fog droplets in the air. While the TEOM also shows an increase, it does not record an increase as high as the PMS1003. As fog begins to form, we observe an increase in both the PNC and $PM_{2.5}$ concentration reported by the PMS1003. The corresponding increase in the TEOM reading, although significantly smaller than the PMS1003, suggests that, in the presence of fog, the dryer at its inlet has a limited efficiency in
terms of removing the liquid phase of the particles.

An obvious question that arises from this work is whether it is possible to derive a correction factor for the particle number and mass concentrations reported by the low-cost sensors in the presence of high humidity and fog. Our results show that, once the deliquescence point is exceeded, the particle number and mass concentrations begin to increase and are not directly related to the absolute value of the relative humidity. Once the ambient temperature reaches the dew point temperature, the conditions become suitable for the formation of fog droplets in the air and, since a significant fraction of these water droplets fall within the detection size of the PMS1003 (Fig 7), they are detected as particles. We also observed that the PNC and PM concentrations reported by the PMS1003 decreased in the presence of rain. This is not unexpected as it is known that rain washes out a fraction of airborne particles. More interestingly, our results show that the decrease in PNC and PM concentrations reported by the PMS1003 due to rain were significantly greater when there was an episode of fog than when there was no fog. While a significant number of fog droplets fall within the detection size range of the PMS1003, almost all the rain drops are larger than the maximum detection size of particles. We hypothesize that the raindrops were washing out the fog droplets in the air, resulting in an overall decrease in the reported PNC and PM concentrations reported by the low-cost particle sensors that have no drying facilities at their sample inlets. Moreover, the relative humidity of the atmosphere increased during rain, often approaching 100%. Raindrops are too large to be detected by most particle sensors and, as such, they do not show an increase in concentration during rain. For these reasons, we find that there is no direct relationship between the relative humidity in the atmosphere and the PNC and PM concentrations reported by a sensor or monitor with no drying facility at its inlet and, as such, it is not possible to derive any appropriate correction factors for this effect.

Since they generally do not have drying facilities at their sample inlets, low-cost particle sensors measure what is actually present in the air, including both the solid and liquid phases of the particles. This is a real observation and not an artefact of the instrument as suggested by Crilley et al. (2018). This is an important aspect to be kept in mind when using low-cost sensors to assess the pollution levels in the atmosphere. What this illustrates is that it should not be presumed that low-cost sensors are suited for regulatory applications. For example, while it is reasonable to use low-cost sensors to measure the actual particle mass concentrations that are present in the air; such observations should not be used to verify if the air quality meets the stipulated guidelines or standards for particle pollution.

**Acknowledgements**

We would like to thank the Queensland Department of the Environment and Science for providing the facilities and data from the air quality monitoring station. This study was supported by Linkage Grant LP160100051 from the Australian Research Council. We are grateful for useful discussions with Graham Johnson and Gavin Fisher. Akwasi Asumadu-Sakyi, Mawutorli Nyarku and Riki Lamont assisted with the field work.

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

20

**Tables**

Table to be placed under Figure 7

| | | | | |
|---|---|---|---|---|
| Relative Humidity (%) | 85.9 | 75.9 | 61.9 | 36.3 |
| PM$_{2.5}$ PMS1003 ($\mu$g m$^{-3}$) | 41.3 | 57.4 | 29.3 | 16.1 |
| PM$_{2.5}$ TEOM ($\mu$g m$^{-3}$) | 24.3 | 34.9 | 20.8 | 16.5 |

**List of Figure Captions**

Figure 1: Composition of particles in the atmosphere of Brisbane, as derived from Harrison (2007).

Figure 2: The The $PM_{2.5}$ concentration reported by the PMS1003 and the DustTrak as the relative humidity was increased in the laboratory chamber.

Figure 3: Time series of the $PM_{2.5}$ concentrations reported by the PMS1003 and the standard TEOM, together with the relative humidity, during the entire duration of the study.

Figure 4: The hourly $PM_{2.5}$ concentration reported by the PMS1003 and TEOM over a humid night (August 7) at the outdoor monitoring station. The arrows show the changing trends.

Figure 5: Graph of PNC>1.0 µm against the PNC between 0.3-0.5 µm during a day that included a period of fog (July 31). The straight line represents the best fit through the points under the broken line only.

Figure 6: Variation of the $PM_{2.5}$ concentration reported by the PMS1003 and TEOM during a day (a) with no fog (July 24)
and (b) with early morning fog (July 30).

Figure 7: PNCs reported by the PMS1003 in the six size bins at three hourly intervals during a morning with fog (July 30). Fog was observed between 3:00 and 06:30 AM. The table under the figure gives additional information at the respective times.


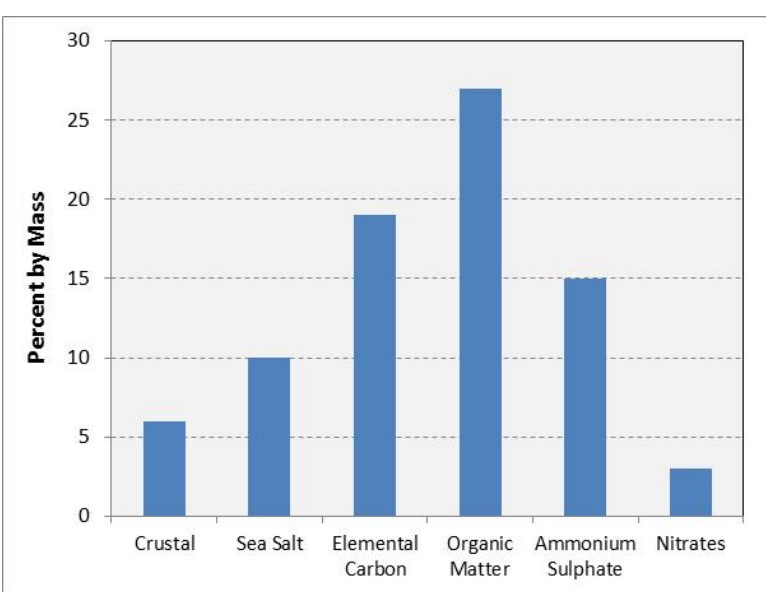

5    **Figure 1: Composition of particles in the atmosphere of Brisbane, as derived from Harrison (2007).**

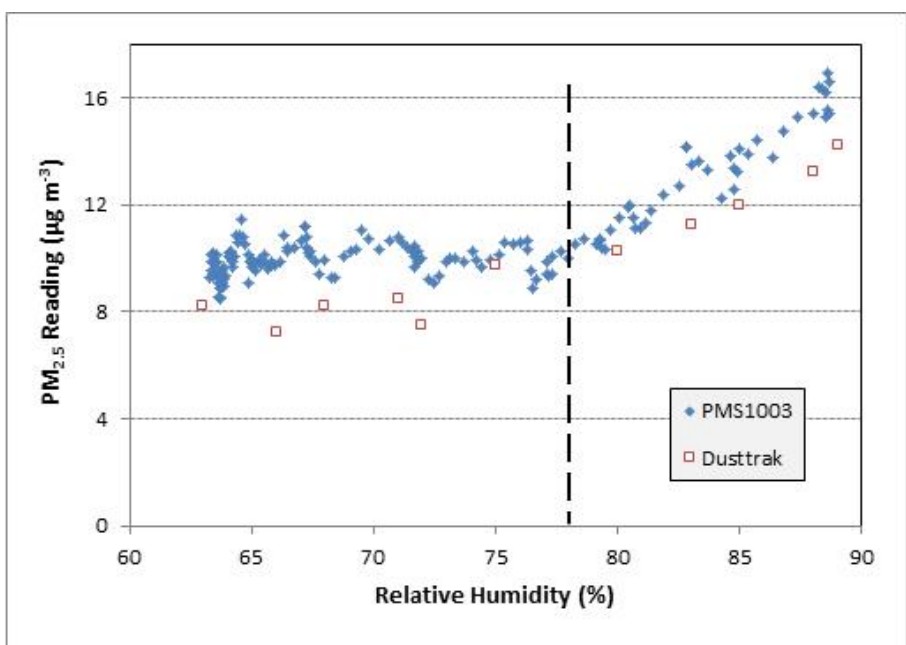

**Figure 2: The The PM$_{2.5}$ concentration reported by the PMS1003 and the DustTrak as the relative humidity was increased in the laboratory chamber.**

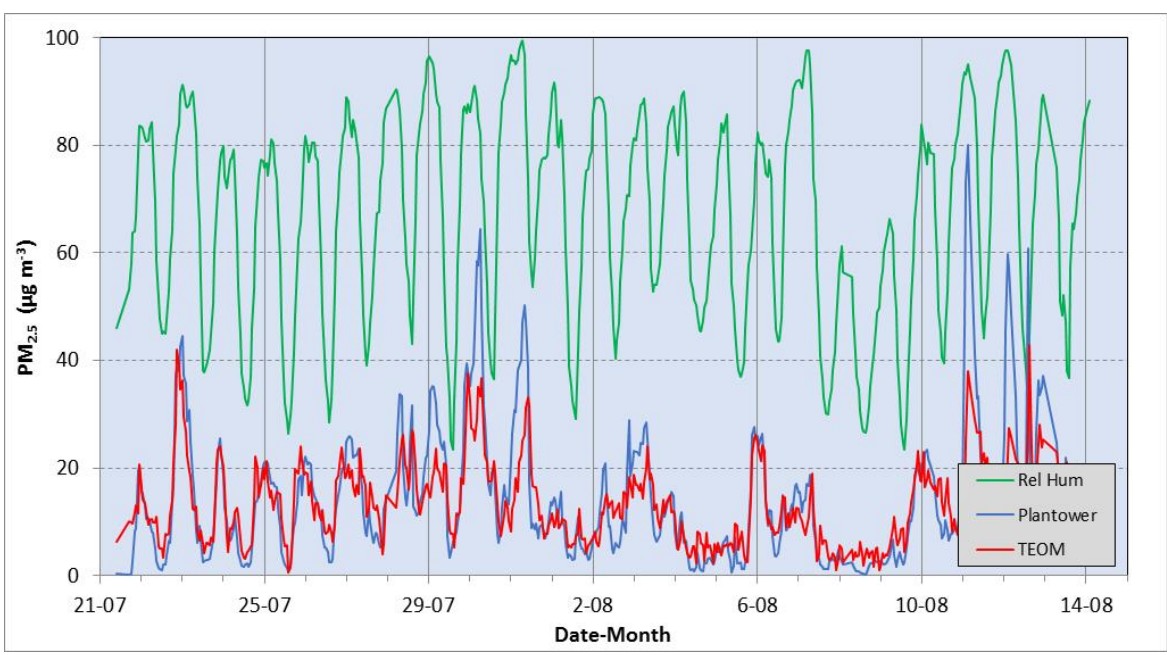

**Figure 3: Time series of the PM$_{2.5}$ concentrations reported by the PMS1003 and the standard TEOM, together with the relative humidity, during the entire duration of the study.**

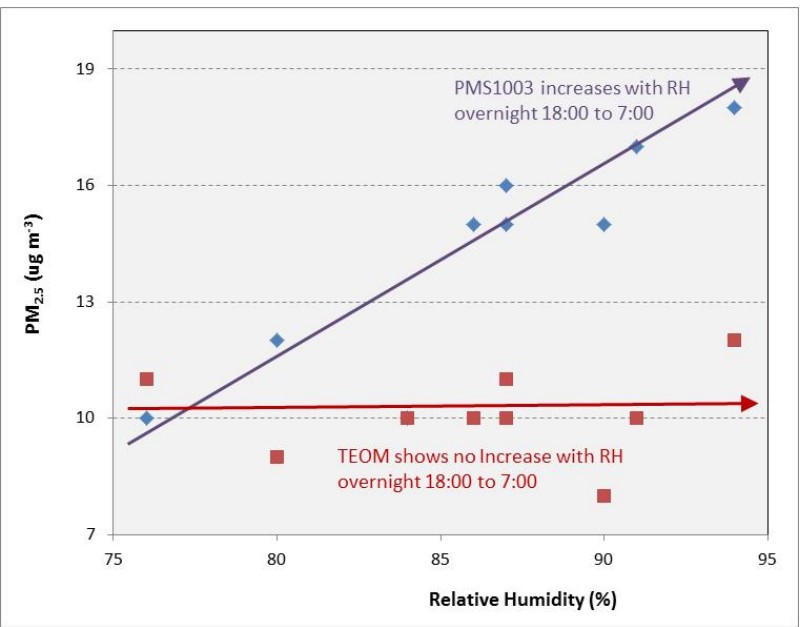

**Figure 4: The hourly PM<sub>2.5</sub> concentration reported by the PMS1003 and TEOM over a humid night (August 7) at the outdoor monitoring station. The arrows show the changing trends.**

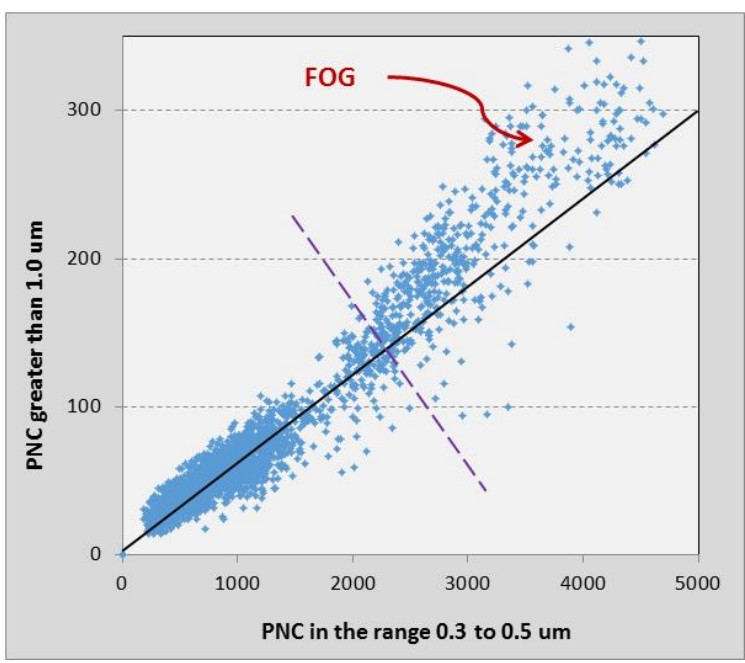

**Figure 5: Graph of PNC>1.0 μm against the PNC between 0.3-0.5 μm during a day that included a period of fog (July 31). The straight line represents the best fit through the points under the broken line only.**

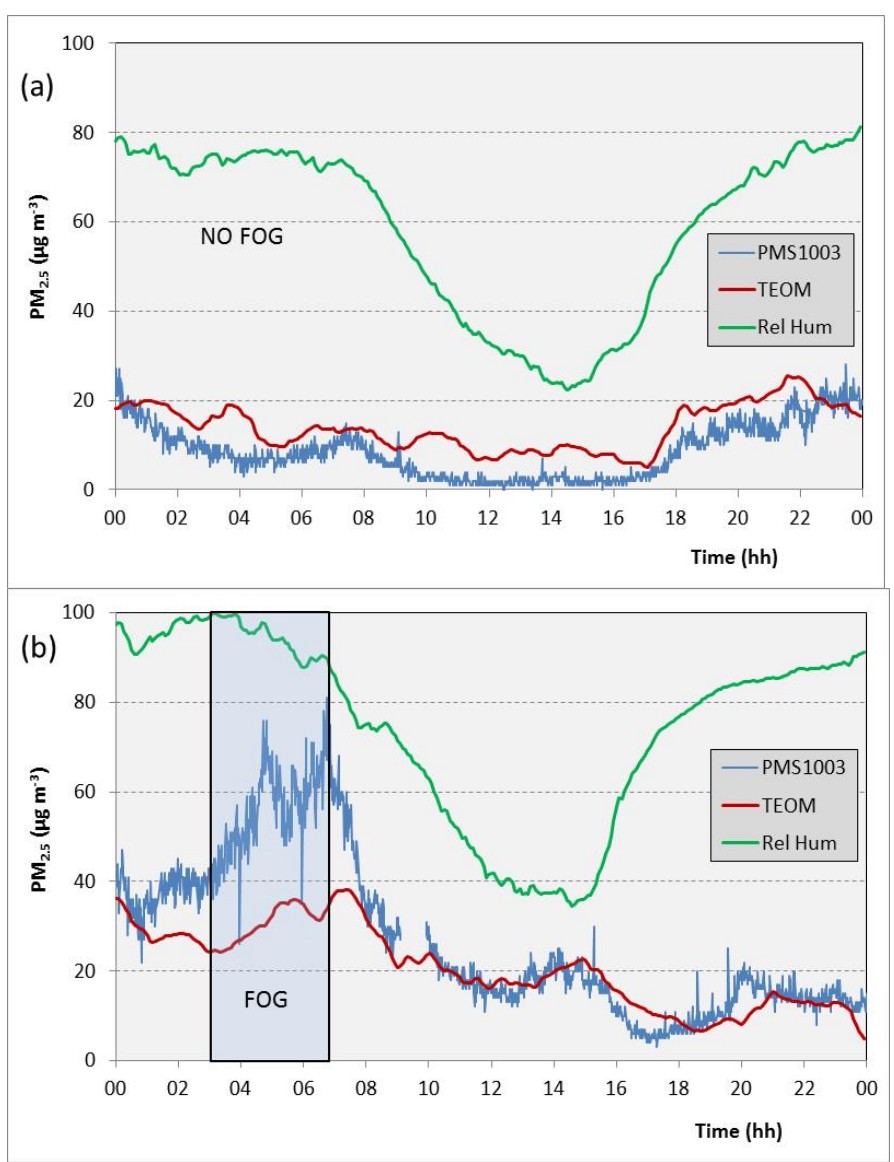

**Figure 6: Variation of the PM2.5 concentration reported by the PMS1003 and TEOM during a day (a) with no fog (July 24) and (b) with early morning fog (July 30).**

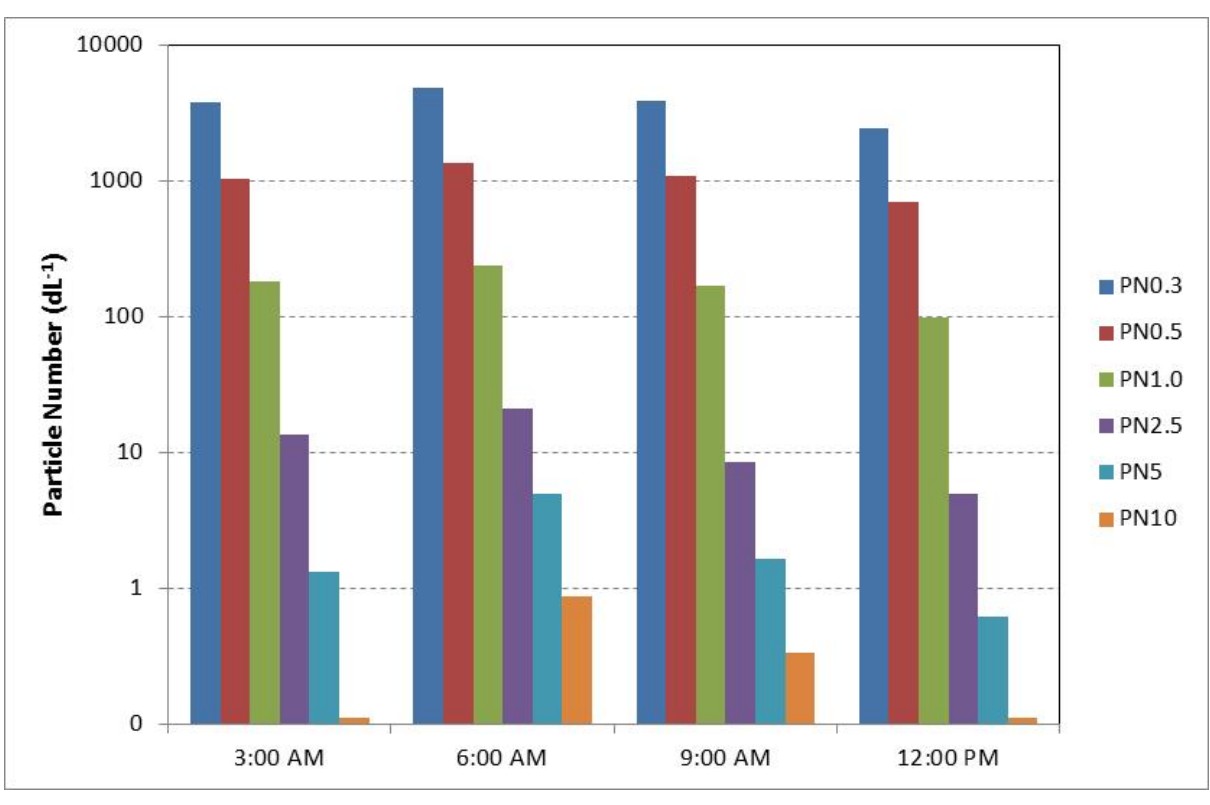

| | 3:00 AM | 6:00 AM | 9:00 AM | 12:00 PM |
|---|---|---|---|---|
| Relative Humidity (%) | 85.9 | 75.9 | 61.9 | 36.3 |
| PM$_{2.5}$ PMS1003 (µg m$^{-3}$) | 41.3 | 57.4 | 29.3 | 16.1 |
| PM$_{2.5}$ TEOM (µg m$^{-3}$) | 24.3 | 34.9 | 20.8 | 16.5 |

**Figure 7: PNCs reported by the PMS1003 in the six size bins at three hourly intervals during a morning with fog (July 30). Fog was observed between 3:00 and 06:30 AM. The table under the figure gives additional information at the respective times.**

