# Peer review of "The Influence of Humidity on the Performance of Low-Cost Air Particle Mass Sensors and the Effect of Atmospheric Fog"

_Atmospheric Measurement Techniques, 2018_

## Referee Comment (RC1) · Anonymous Referee #1 · 10 May 2018

This manuscript presents an interesting assessment of the influence of relative humidity on the performance of one low cost sensor. The results are robust, even if the amount of data presented could be considered scarce. They are useful in general for the scientific community. I would favor publication, but a number of relevant issues should be addressed first:

Title: please modify to "the performance of a low cost sensor", as the authors mainly analyze one type of sensor and the title is therefore misleading. The few data presented for another 4 sensors do not justify generalizing in the title.

Page 1: Line 19: "sensors can accurately report particle mass and number concentra-

tions", please remove as this is not a conclusion from this work. The authors have not studied the overall performance of sensors. Line 30, reference needed.

Page 2: References needed in lines 3, 4 and 5. In general, please review references in the introduction, as they are scarce Line 7: "the performance of low cost" should be "the performance of one low cost"

Page 3: Line 23: "out of each", sentence unfinished Line 29: what were the results from the intercomparison of the Dusttrak? They could be useful, even if in Supporting Information

Page 4: Line 29: how do the authors know? The Dusttrak concentrations also increased with RH>78%. Is 1.8 the ration between the sensor and the Dusttrak readings? If so, what was the ratio for RH between 60-75%? Please clarify these issues: with the data in figure 1 it is not possible for the reader to extract the conclusions in lines 28-30 on page 4

Page 5: Line 1: could the PNC time series be added to figure 1? It would be interesting to see Lines 9-14: these are not original results from the authors and should be moved to the introduction. Especially, figure 2 should be removed as it is published material and in addition it doesn't add relevant information for the paper. Line 31: "illustrating" should be "suggesting". This comparison is useful, but it can only suggest. Without a collocated CPC it is not possible to conclude firmly that the increase in PNC is an artifact and not that an additional source could be present which by chance correlates with foggy scenarios. However unlikely this is, it can't be ruled out with the data presented by the authors

Figure 7: same as for figure 2, it should be removed as it is not primary research by the authors (it is already published by other authors). Therefore it should be removed and, if anything, referenced in the introduction

Page 7, lines 15-16: why is this an "interesting observation", if it is "not unexpected" by

the authors? It all seems quite expectable (until line 22) page 7, line 33: "sensors are not always fit for purpose", I believe there is a misconception here: precisely because they are fit for purpose they shouldn't be used to verify compliance with standards, as this is not the purpose that sensors are designed for. This should be the message to be conveyed, to that they are not fit for purpose.

---

## Referee Comment (RC2) · Anonymous Referee #2 · 1 Jul 2018

This paper has the possibility of being a good paper, with some considerable extra additional text and analysis. The science area being discussed is very topical and the authors seek to show a particle-bound water effect on the DustTrak instrument measurements compared to the dried TEOM data.

Some specific areas which I think could improve the manuscript:

1.Referencing

The authors only cite 15 papers, despite there being a significant body of work now in this area. For example, Kingham et al. https://www.sciencedirect.com/science/article/pii/S1352231005008885 who Specifically compare TEOMs with dusttrak and other instruments (And cites other measurement which have shown the differences and over reading of the instruments which do not dry the particles. I could have chosen several other papers.

The authors should do a more detailed literature review and clearly define the reasons why their new study adds to the wider knowledge

2. Quality assurance of laboratory experiment

The description of the lab results is short and not detailed enough to have confidence in the work. The chamber at 1 m3 is rather small and probably has significant wall effects from particle deposition/emission. No description of how this is checked for is reported - the citation to the previous study is not enough for a reader to have confidence in the methodology. Only one set of experimental results are shown. Ideally at least 5 repeats with similar PM loading should be shown with an uncertainty analysis/error bars etc. In addition a blank run to show there is no effect of background in the chamber.

Did the authors do any runs with laboratory derived aerosol?

3. Field study.

Despite running the experiment for a month nearly the authors only show the one day (when there was fog). In the results they talk about several different periods but it is hard to have a clear overview as no dates are mentioned. It would be better to show the full dataset along with the associated local meteorology. Discuss that larger dataset and then focus in on the fog events. Did the RH get close to 100 in the absence of fog? Did the over reading occur then?

The authors do not mention the systematic under reading by the dustrak cf the TEOM in the non fog part of the graphs shown. No summary statistics of the comparisons between the instruments over the intercomparison period shown - Which I think is essential to understand the bias and offsets. It would be good to see the statistics of different RH bands.

4. Fog vs high RH:

Though ther is some discussion about the process of fog formation and particle activation, the fog effect is mixed up with deliquescence. What the DustTrak observes is aqueous particles, and the TEOMs will have a mixture of effloresced and supersaturated aqueous particles. The growth curve either from a dry or supersaturated aqueous particle is similar. This is completely different to fog formation which is cloud particle activation which occurs when RH>100%. The author should clearly separate these two regimes and these separate processes, both in the results and discussion.

Though I am reasonably sure the observation of fog and rain with the different measurements by both instruments are correct, and are somewhat explained by the authors, there is a more scientific discussion required to make sense of the results. It would also be useful to see a clear extrapolation to more a general discussion about what type of fog it was (see https://www.atmos-them-phys.net/14/10517/2014/acp-14-10517-2014.pdf for definitions) and how typical the short period of observation they have is (I.e. how would it affect annual data capture, could you write a met station interface to remove data with fog?

Figure 2 and 7 are not needed as these are well know in the literature, and only meet mentioning in the text.

Following a major re-write it is likely that further comments would arise.

---

## Author Comment (AC1) · 24 Jul 2018

Overall Comments This manuscript presents an interesting assessment of the influence of relative humidity on the performance of one low cost sensor. The results are robust, even if the amount of data presented could be considered scarce. They are useful in general for the scientific community. I would favor publication, but a number of relevant issues should be addressed first:

Comment 1 Title: please modify to "the performance of a low cost sensor", as the authors mainly analyze one type of sensor and the title is therefore misleading. The few data presented for another 4 sensors do not justify generalizing in the title.

[Figure]

Response 1 We have amended the title as suggested.

Comment 2 Page 1: Line 19: "sensors can accurately report particle mass and number concentrations", please remove as this is not a conclusion from this work. The authors have not studied the overall performance of sensors.

Response 2 We have removed this text and amended this sentence as follows:

"This study shows that it is important to understand that the results provided by low-cost particle sensors, such as the PMS1003, cannot be used to ascertain if air quality standards are being met".

Comment 3 Line 30, reference needed.

Response 3 The following references have been inserted: Snyder et al., 2013; Jovasevic-Stojanovic et al., 2015; Rai et al., 2017.

Comment 4 Page 2: References needed in lines 3, 4 and 5. In general, please review references in the introduction, as they are scarce.

Response 4 The following references have been inserted: Jayaratne et al., 2018; Rai et al., 2017; Kelly et al., 2017. Fine et al., 2010; Piedrahita et al., 2014. Holstius et al., 2014; Rai et al., 2017; Crilley et al., 2018; Jayaratne et al., 2018.

Comment 5 Line 7: "the performance of low cost" should be "the performance of one low cost"

Response 5 The sentence has been changed to "... the performance of a low-cost particulate matter sensor".

Comment 6 Page 3: Line 23: "out of each", sentence unfinished

Response 6 Text changed to "...airflow into and out of the device".

Comment 7 Line 29: what were the results from the intercomparison of the Dusttrak? They could be useful, even if in Supporting Information

Response 7 These results are detailed in our publication Jayaratne et al., 2018. In this paper, we have inserted the following text:

"Prior to the study, the DustTrak was calibrated against a standard TEOM in the laboratory. With dry ambient aerosols, the PM2.5 concentrations reported by the two instruments agreed to within 10% (Jayaratne et al., 2018). With normal ambient aerosols, the readings again agreed closely until the relative humidity exceeded about 75% when the DustTrak readings were significantly greater than that of the TEOM".

Comment 8 Page 4: Line 29: how do the authors know? The Dusttrak concentrations also increased with RH>78%. Is 1.8 the ration between the sensor and the Dusttrak readings? If so, what was the ratio for RH between 60-75%? Please clarify these issues: with the data in figure 1 it is not possible for the reader to extract the conclusions in lines 28-30 on page 4

Response 8 The deviation is from the concentration value at lower RH. We have changed the text to:

"The critical relative humidity beyond which the PM2.5 concentration reported by the PMS1003 begins to deviate from the previous ambient value is indicated by the broken line in the figure".

No. This refers to the increase in the PMS1003 reading from its steady value at low RH values. There is a 80% increase between a RH of 75% and 88%, hence the ratio 1.8. Accordingly, there is no increase between 60% and 75%, so the ratio here will be 1. We have changed the text as follows:

"Beyond this value, the PM2.5 readings indicated by the PMS1003 increased steadily from about 9 $\mu$g m-3 at a relative humidity of 78% to about 16 $\mu$g m-3 at the maximum relative humidity of 89% achieved in this experiment – an increase of almost 80%".

Comment 9 Lines 9-14: these are not original results from the authors and should be moved to the introduction. Especially, figure 2 should be removed as it is published

material and in addition it doesn't add relevant information for the paper.

Response 9 The material is relevant as it shows that there are many deliquescent substances in the Brisbane environment which is used to explain our results. Also, this Figure is not directly from the paper cited (Harrison, 2007) but we have plotted it based on data provided in that paper. However, as suggested, we have moved the figure (now labelled 'Fig 1') and the following text to the Introduction:

"The composition of particles in the atmosphere of Brisbane, as derived from Harrison (2007) is shown in Fig. 1. The subtropical, near-coastal environment is characterised by the presence of several hygroscopic salts such as sodium chloride, ammonium sulphate and ammonium nitrate that have deliquescence relative humidities in the range of 70% to 80% (Hu et al., 2010). Many particles in the air in Brisbane contain these salts in varying concentrations. Once the relative humidity exceed the respective deliquescence values, those salts begin to absorb water, resulting in particle growth and the excess water is registered by PM sensors, unless they are removed at the instrument inlets by heating or drying. While more expensive instruments such as the TEOM have built-in drying features at the sample inlets, it is not standard on low-cost sensors and even in many other mid-cost monitors such as the TSI DustTrak".

Comment 10 Line 31: "illustrating" should be "suggesting". This comparison is useful, but it can only suggest. Without a collocated CPC it is not possible to conclude firmly that the increase in PNC is an artifact and not that an additional source could be present which by chance correlates with foggy scenarios. However unlikely this is, it can't be ruled out with the data presented by the authors.

Response 10 We agree and have replaced the word "illustrating" with "suggesting".

Comment 11 Figure 7: same as for figure 2, it should be removed as it is not primary research by the authors (it is already published by other authors). Therefore it should be removed and, if anything, referenced in the introduction

Response 11 As suggested, we have removed Figure 7 from the paper.

Comment 12 Page 7, lines 15-16: why is this an "interesting observation", if it is "not unexpected" by the authors? It all seems quite expectable (until line 22) page 7, line 33: "sensors are not always fit for purpose", I believe there is a misconception here: precisely because they are fit for purpose they shouldn't be used to verify compliance with standards, as this is not the purpose that sensors are designed for. This should be the message to be conveyed, to that they are not fit for purpose.

Response 12 We agree. We have deleted this text and revised the two sentences as follows:

Page 7, lines 15-16: "We also observed that the PNC and PM concentrations reported by the PMS1003 decreased in the presence of rain".

Page 7, line 33: "What this illustrates is that it should not be presumed that low-cost sensors are suited for regulatory applications".

---

## Author Comment (AC2) · 24 Jul 2018

Overall Comments This paper has the possibility of being a good paper, with some considerable extra additional text and analysis. The science area being discussed is very topical and the authors seek to show a particle-bound water effect on the DustTrak instrument measurements compared to the dried TEOM data.

Some specific areas which I think could improve the manuscript:

Comment 1 1.Referencing The authors only cite 15 papers, despite there being a significant body of work now in this area. For example, Kingham et

al. https://www.sciencedirect.com/science/article/pii/S1352231005008885 who Specifically compare TEOMs with dusttrak and other instruments (And cites other measurement which have shown the differences and over reading of the instruments which do not dry the particles. I could have chosen several other papers.

Response 1 We have cited the following two additional papers in the Introduction:

Kingham, S., Durand, M., Aberkane, T., Harrison, J., Wilson, J.G., Epton, M., Winter comparison of TEOM, MiniVol and DustTrak PM10 monitors in a woodsmoke environment, Atmospheric Environment, 40(2), 338-347, 2006.

Alexandrova, O.A., Boyer, D.L., Anderson, J.R., Fernando, H.J.S., The influence of thermally driven circulation on PM10 concentration in the Salt Lake Valley, Atmospheric Environment, 37(3), 421-437, 2003.

Including these, seven new papers have been cited in this paper now.

Comment 2 The authors should do a more detailed literature review and clearly define the reasons why their new study adds to the wider knowledge

Response 2 We have done a more detailed literature review that has added seven more citations into this paper. The contribution of this paper to wider knowledge is now reflected in the following text:

At the end of the Introduction:

"While several studies have drawn attention to a possible effect of humidity on the performance of low cost sensors, no study has reliably quantified the effect. This study was carried out to investigate and to assess the magnitude of the effect of relative humidity on the performance of a low-cost particle sensor and to understand the mechanisms involved".

The last paragraph in the Discussion section:

"Since they generally do not have drying facilities at their sample inlets, low-cost particle sensors measure what is actually present in the air, including both the solid and liquid phases of the particles. This is a real observation and not an artefact of the instrument as suggested by Crilley et al. (2018). This is an important aspect to be kept in mind when using low-cost sensors to assess the pollution levels in the atmosphere. What this illustrates is that it should not be presumed that low-cost sensors are suited for regulatory applications. For example, while it is reasonable to use low-cost sensors to measure the actual particle mass concentrations that are present in the air; such observations should not be used to verify if the air quality meets the stipulated guidelines or standards for particle pollution".

Comment 3 2. Quality assurance of laboratory experiment The description of the lab results is short and not detailed enough to have confidence in the work. The chamber at 1 m3 is rather small and probably has significant wall effects from particle deposition/emission. No description of how this is checked for is reported - the citation to the previous study is not enough for a reader to have confidence in the methodology. Only one set of experimental results are shown. Ideally at least 5 repeats with similar PM loading should be shown with an uncertainty analysis/error bars etc. In addition a blank run to show there is no effect of background in the chamber. Did the authors do any runs with laboratory derived aerosol?

Response 3 The focus of this paper is not on lab studies but on the field studies because many air quality stations use standard instruments such as the TEOM and BAM that have drying facilities to remove any water in the sample. This is appropriate as air quality standards are based on the solid fraction of the aerosols only. The main conclusion of this study is that low cost sensors that are being increasingly used in field studies today, do not have a sample drying facility and the PM values reported will include any liquid fraction and therefore should not be used for regulatory purposes. The laboratory experiments, including those carried out with five different low cost sensors, four different aerosols, at a range of relative humidity and temperature, are described in detail in our earlier publication (Jayaratne et al., 2018; Plos One) and anyone who

is interested in the details can read it in there. In this paper, we merely mention the laboratory experiments to show that the reported PM2.5 values are affected at relative humidities above about 75%. This observation leads to the more detailed field studies under real world humid conditions and episodes of fog that we focus on in this paper.

Comment 4 3. Field study. Despite running the experiment for a month nearly the authors only show the one day (when there was fog). In the results they talk about several different periods but it is hard to have a clear overview as no dates are mentioned. It would be better to show the full dataset along with the associated local meteorology. Discuss that larger dataset and then focus in on the fog events. Did the RH get close to 100 in the absence of fog? Did the over reading occur then?

Response 4 We have now presented the data from the entire period of observation (new Fig 3). The dates on the x-axis are shown from 21st July to 14th August 2017. The respective dates are also included in the captions of the other figures. The figure includes the time series of the relative humidity.

A description and discussion of the larger dataset is provided in the following text that has been inserted on page 6:

"Fig 3 shows the time series of the PM2.5 concentrations reported by the PMS1003 and the standard TEOM during the entire duration of the study. Also shown is the relative humidity during this period. The relative humidity exhibited a daily cycle with a minimum in the early afternoon and a maximum at night. Note that the peak PM2.5 concentrations indicated by both instruments generally coincided with the time when the relative humidity reached its maximum value near dawn each day. The maximum value often coincided with episodes of fog, although its value did not reach 100%. It is likely that this was a consequence of a limitation of the instrument. At such times, the PMS1003 reading was generally higher than the TEOM. However, from Fig 3, it is observed that on many days, the readings of both instruments increased during times when the relative humidity was high, suggestive that the TEOM did not remove all of

the liquid portion of the aerosols. In the afternoon, the TEOM reading was often higher than the PMS1003. This is probably because most of the aerosols in the atmosphere at this time were ultrafine particles from motor vehicle emissions. The size of these particles are below the minimum detectable size limit of the PMS1003 which is 0.3 $\mu$m".

Comment 5 The authors do not mention the systematic under reading by the dustrak cf the TEOM in the non fog part of the graphs shown. No summary statistics of the comparisons between the instruments over the intercomparison period shown - Which I think is essential to understand the bias and offsets. It would be good to see the statistics of different RH bands.

Response 5 We assume that the Reviewer is referring to Fig 2, as this is the only graph that shows data from the DustTrak particle monitor. In general, the readings of the PMS1003 sensors differed between the individual units. The differences depended on the type of aerosol and the concentration being measured. At the low concentrations found in the ambient environment of Brisbane, the coefficient of variation between 'identical' PMS1003 sensor units was about 0.07 and this is reported in detail in Jayaratne et al. (2018).

As observed in Fig 2, the PM2.5 readings of the particular PMS1003 sensor used in this experiment were about 20% higher than the readings on the DustTrak. In the present study, the absolute magnitudes of the PM2.5 concentrations are not considered to be important since the paper is focussed on the differences in PM2.5 concentrations reported by the DustTrak and the PMS1003 sensors as the relative humidity varies. We show that the concentration remains relatively constant up to a relative humidity of about 75% and then rises sharply as it is increased further. It is the difference in the reported concentrations that is important, not their actual magnitudes.

However, we have inserted the following text in the description of the graph in Fig 2 in Section 3.1:

"As observed in the figure, the PM2.5 readings of the particular PMS1003 sensor used in this experiment were consistently higher than the readings on the DustTrak. In general, the readings of the PMS1003 sensors differed between the individual units. The differences depended on the type of aerosol and the concentration being measured. At the low concentrations found in the ambient environment of Brisbane, the coefficient of variation between 'identical' PMS1003 sensor units was about 0.07, as reported in detail in Jayaratne et al. (2018)".

Comment 6 4. Fog vs high RH: Though ther is some discussion about the process of fog formation and particle activation, the fog effect is mixed up with deliquescence. What the DustTrak observes is aqueous particles, and the TEOMs will have a mixture of effloresced and supersaturated aqueous particles. The growth curve either from a dry or supersaturated aqueous particle is similar. This is completely different to fog formation which is cloud particle activation which occurs when RH>100%. The author should clearly separate these two regimes and these separate processes, both in the results and discussion.

Though I am reasonably sure the observation of fog and rain with the different measurements by both instruments are correct, and are somewhat explained by the authors, there is a more scientific discussion required to make sense of the results. It would also be useful to see a clear extrapolation to more a general discussion about what type of fog it was (see https://www.atmos-them-phys.net/14/10517/2014/acp-14-10517-2014.pdf for definitions) and how typical the short period of observation they have is (I.e. how would it affect annual data capture, could you write a met station interface to remove data with fog?

Response 6 We agree with the reviewer's comments. However, the mechanism of water uptake by particles, the formation of fog in the atmosphere and the physics behind the removal of fog by rain are beyond the scope of this paper. Here, we focus on the effect of wet aerosols and fog droplets in the air on the readings reported by low cost particle sensors and other particle sensors with no drying facilities at the inlet. We did

not have any means of determining whether there was fog at RH<100%. The reports of fog in this study are all based on visual observations and may not be exact when the RH is near 100%. We have inserted the word "visually" before "observed" in at least three sentences in Section 3.2:

"Fog was visually observed at the site during the early morning hours".

"In Fig 5, we show the number concentration of particles larger than 1.0 $\mu$m against the corresponding number in the lowest size bin, 0.3 to 0.5 $\mu$m on the 31st August when there was an episode of fog visually observed during the early morning".

"Fig. 6(a) shows the results on the 24th July when the relative humidity did not exceed 80% and there were no visual reports of fog".

Regarding removing data with fog, our results show that it is impossible to predict the formation of fog based on the relative humidity alone. This is discussed in detail in the following paragraph in the Results and Discussion Section:

"An obvious question that arises from this work is whether it is possible to derive a correction factor for the particle number and mass concentrations reported by the low-cost sensors in the presence of high humidity and fog. Our results show that, once the deliquescence point is exceeded, the particle number and mass concentrations begin to increase and are not directly related to the absolute value of the relative humidity. Once the ambient temperature reaches the dew point temperature, the conditions become suitable for the formation of fog droplets in the air and, since a significant fraction of these water droplets fall within the detection size of the PMS1003 (Fig 7), they are detected as particles. We also observed that the PNC and PM concentrations reported by the PMS1003 decreased in the presence of rain. This is not unexpected as it is known that rain washes out a fraction of airborne particles. More interestingly, our results show that the decrease in PNC and PM concentrations reported by the PMS1003 due to rain were significantly greater when there was an episode of fog than when there was no fog. While a significant number of fog droplets fall within the detection size range of

the PMS1003, almost all the rain drops are larger than the maximum detection size of particles. We hypothesize that the raindrops were washing out the fog droplets in the air, resulting in an overall decrease in the reported PNC and PM concentrations reported by the low-cost particle sensors that have no drying facilities at their sample inlets. Moreover, the relative humidity of the atmosphere increased during rain, often approaching 100%. Raindrops are too large to be detected by most particle sensors and, as such, they do not show an increase in concentration during rain. For these reasons, we find that there is no direct relationship between the relative humidity in the atmosphere and the PNC and PM concentrations reported by a sensor or monitor with no drying facility at its inlet and, as such, it is not possible to derive any appropriate correction factors for this effect".

Comment 7 Figure 2 and 7 are not needed as these are well know in the literature, and only meet mentioning in the text.

Response 7 Also, in response to Reviewer 1, we have moved Fig 2 to the Introduction (as Fig 1 now) and, as suggested, we have removed Fig 7 from the paper.